# Iatrogenic Atrial Septal Defect after Intracardiac Echocardiography-Guided Left Atrial Appendage Closure: Incidence, Size, and Clinical Outcomes

**DOI:** 10.3390/jcm12010160

**Published:** 2022-12-25

**Authors:** Yibo Ma, Lanyan Guo, Jie Li, Haitao Liu, Jian Xu, Hui Du, Yi Wang, Huihui Li, Fu Yi

**Affiliations:** Department of Cardiology, Xijing Hospital, Air Force Medical University, Xi’an 710032, China

**Keywords:** iatrogenic atrial septal defect, intracardiac echocardiography, left atrial appendage closure, atrial fibrillation, right-to-left shunt

## Abstract

Background: The data on iatrogenic atrial septal defect (iASD) after left atrial appendage closure (LAAC), especially intracardiac echocardiography (ICE)-guided LAAC, are limited. Compared with transesophageal echocardiography (TEE)- or digital subtraction angiography (DSA)-guided LAAC, the transseptal puncture (TP) ICE-guided LAAC is more complicated. Whether or not ICE-guided TP increases the chances of iASD is controversial. We investigate the incidence, size, and clinical outcomes of iASD after ICE-guided LAAC. Methods: A total of 177 patients who underwent LAAC were enrolled in this study and were assigned to the ICE-guided group (group 1) and the TEE- or DSA-guided group (group 2). Echocardiography results and clinical performances at months 2 and 12 post-procedure were collected from the electronic outpatient records. Results: A total of 112 and 65 patients were assigned to group 1 and group 2, respectively. The incidence of iASD at follow-up (FU) month 2 was comparable between the groups (21.4% in group 1 vs. 15.4% in group 2, *p* = 0.429). At month 12 of FU, the closure rate of iASD was comparable to that of group 2 (70.6% vs. 71.4%, *p* = 1.000). No right-to-left (RL) shunt was observed among the iASD patients during the FU. Numerically larger iASD were observed in group 1 patients at month 2 of FU (2.8 ± 0.9 mm vs. 2.2 ± 0.8 mm, *p* = 0.065). No new-onset of pulmonary hypertension and iASD-related adverse events were observed. Univariable and multivariable logistic regression analysis showed that ICE-guided LAAC was not associated with the development of iASD (adjusted OR = 1.681; 95%CI, 0.634–4.455; *p* = 0.296). Conclusions: The ICE-guided LAAC procedure does not increase the risk of iASD. Despite the numerically large size of the iASD, it did not increase the risk of developing adverse complications.

## 1. Introduction

Transcatheter left atrial appendage closure (LAAC) has become an effective and safe alternative to oral anticoagulant (OAC) in patients at high risk of atrial fibrillation (AF) and anticoagulation intolerance [1,2,3]. Transseptal puncture (TP) is performed during the LAAC procedure, providing access for a 12-F closure device delivery sheath to the left atrial appendage (LAA) through the left atrium. The fossa ovalis, the weakest part of the atrial septum, was the most common site for TP in the past. Because acute transseptal shunts were detected in almost all patients post-procedure, this site is currently seldom considered [4]. The most common site for TP is now the posterior and inferior area of the fossa ovalis, which is associated with a lower incidence of immediate left-to-right (LR) shunting and allows for more accurate delivery of the sheath [5]. The reliability of the puncture of the posterior and inferior area of the fossa ovalis is higher than that of the puncture of the fossa ovalis.

Transesophageal echocardiography (TEE), intracardiac echocardiography (ICE), and digital subtraction angiography (DSA) were the most commonly used methods for guiding the LAAC procedure. Using ICE to deliver a catheter into the left atrium during the LAAC has unique advantages [6,7]. For instance, compared with the TEE method, it is feasible and safe, and general anesthesia is not needed, which eliminates the usage of contrast, shortening the time in the catheterization room [8]. During the ICE-guided LAAC procedure, more TP operations are performed to construct another access. Whether or not this method can increase the incidence of the iatrogenic atrial septal defect (iASD) remains controversial [8,9]. In addition, a previous report showed that the iASD, especially the right-to-left (RL) shunt iASD, causes acute anoxia and exacerbates pulmonary hypertension in patients who undergo the MitraClip procedure [10]. Considering that most patients who undergo LAAC also have complex underlying complications, the clinical relevance of iASD caused by ICE-guided LAAC should be clarified. Our study compared the incidence, size, and clinical outcomes of iASD after ICE-guided LAAC procedure and TEE- or DSA-guided LAAC.

## 2. Materials and Methods

### 2.1. Study Population

This was a single-center retrospective study. AF patients who underwent LAAC at the First Affiliated Hospital of Air Force Military Medical University, Cardiology Department, between March 2019 and December 2021 were enrolled in this study. The indication of LAAC was according to the local guidelines, which can be summarized as patients who are unwilling to receive long-term OAC, have intolerance to anticoagulation, or are at high risk of embolism. The inclusion criteria were: (1) aged 18–85 years old; (2) non-valvular AF confirmed by both echocardiography and standard 12-lead electrocardiogram; and (3) finished at least 2 months of outpatient follow-up (FU). The exclusion criteria included (1) atrial septal defect confirmed by transthoracic echocardiography (TTE), which had then not been corrected by whichever procedure by the time of the LAAC procedure; (2) history of cardiac surgery; and (3) a thicker sheath that advanced into the left atrium at the time of such LAAC procedure, such as cryoballoon ablation (CBA) combined with LAAC. The patients were assigned to the ICE-guided group (group 1) and the TEE- or DSA-guided group (group 2) based on the type of imaging used. The details of the patient enrollment process are shown in Figure 1. The protocol for this study was approved by the ethics board of the First Affiliated Hospital of Air Force Military Medical University. Written informed consent was acquired from all the patients before the procedure.

### 2.2. Procedure

#### 2.2.1. ICE-Guided TP Procedure

The ICE-guided LAAC was performed as previously described [11]. The intracardiac images for reconstructing the three-dimensional left atrium model were obtained using a 10-F SoundStar ICE catheter (Biosense Webster). Briefly, the right femoral access was first constructed, and an 11-F sheath for advancing the ICE catheter was inserted into the middle of the right atrium. The catheter was rotated until the atrial septal was fully visible. Finally, the posterior and inferior areas of the fossa ovalis were punctured. The delivery device to the sheath was advanced to the left atrium through the punctured hole. In addition, during the LAAC procedure, the ICE probe was advanced into the left atrium through the same atrial septal puncture hole. The WATCHMAN device (Boston Scientific) was used for the procedure. If the LAA morphology was unsuitable for the WATCHMAN device implantation, the LACbes device (PushMed, Shanghai, China) was instead implanted. Both sheaths have a diameter of 12-F. The diagrammatic flow of these steps is shown in Figure 2.

#### 2.2.2. TEE- or DSA-Guided TP Procedure

The TEE- or DSA-guided TP procedure was performed as previously described [5,6,7,8]. Briefly, a single femoral vein access was first constructed. A posterior and inferior TP was attempted under TEE or DSA guidance. Throughout the LAAC procedure, only the device delivery sheath was advanced to the left atrium through the punctured hole. During the TEE-guided TP procedure, the patients were given general anesthesia and intubated when the TEE probe was introduced. The principle of the closure device use was the same as that for the ICE-guided TP procedure.

### 2.3. Outpatient Follow-Up

Postimplant FU was scheduled at 2, 6, and 12 months post-procedure. The iASD results were collected at months 2 and 12. The presence and size of the iASDs were detected and measured using three-dimensional TEE, and the blood flow direction through the iASD was detected by color Doppler ultrasonography. The hemodynamic parameters, especially the pulmonary artery pressure, were also assessed by echocardiography. TTE was used as an alternative to TEE for the iASD examination.

In addition to the echocardiography data, the incidence of clinical adverse events, including cardiovascular death, ischemic stroke, migraine, and readmission for acute heart failure, was also captured during each FU.

### 2.4. Statistical Analysis

Continuous normally distributed variables were presented as mean ± SD. Differences between the groups were analyzed by one-way ANOVA. Non-normally distributed data were expressed as the median and interquartile range (IQR), and corresponding groups were compared by the Mann–Whitney U test. Categorical variables were expressed as count (percentage) and compared by Fisher’s exact test.

The unadjusted risk ratio of each variable was estimated by univariable logistic regression analysis. The relationship between ICE-guided LAAC and the risk of iASD was analyzed by multivariable logistic regression analysis after adjustment for several covariates: the CHA_2_DS_2_-VASc score and the HASBLED score (model 1); age, sex, body mass index (BMI), non-paroxysmal AF, AF duration, heart failure, hypertension, diabetes mellitus, prior ischemic stroke, coronary heart disease, left atrial diameter (LAD), left ventricular ejection fraction (LVEF), pathological mitral regurgitation and tricuspid regurgitation, patent foramen ovale (PFO) occlusion, and the use of different closure devices (model 2). Statistical significance was set at *p* ≤ 0.05. Statistical analysis was performed using IBM SPSS software, version 26.0 (IBM SPSS Inc., Chicago, IL, USA), and R software, version 4.2.0 (Ross Ihaka and Robert Gentleman, Auckland, New Zealand).

## 3. Results

### 3.1. Patient Characteristics at Baseline

A total of 177 patients were enrolled in this study (Figure 1). Among them, 112 were classified in group 1, and 65 patients were classified in group 2. The baseline characteristics of the study patients are summarized in Table 1. The mean age, CHA_2_DS_2_-VASc score, and HASBLED score were 64.5 ± 9.1 years, 3.4 ± 1.8, and 1.8 ± 1.1, respectively. Moreover, 65.5% of the study participants were males. Except for non-paroxysmal AF and pulmonary hypertension, which were significantly higher in the patients in group 2 (non-paroxysmal AF: 61.6% vs. 83.1%, *p* = 0.004 and pulmonary hypertension: 6.3% vs. 24.6%, *p* = 0.001), there was no significant difference in patient characteristics at the baseline between the two groups.

### 3.2. Procedure Characteristics

All the patients underwent successful TP. For group 2, 37 (56.9%) patients underwent TEE-guided LAAC, and 28 (43.1%) patients underwent DSA-guided LAAC. The successful LAAC rate in group 1 was 99.1%, and it was 98.5% for the group 2 patients. Significantly more patients in group 1 underwent radiofrequency catheter ablation (RFCA) at the time of LAAC (90.2% vs. 40.0%, *p* = 0.000). All nine patients who underwent PFO occlusion were in group 2. After discharge, all the patients received OAC or antiplatelet therapy (Table 2).

### 3.3. Two Months FU

At a median of 57 (IQR: 47, 74) days of FU, 24 (21.4%) patients in group 1 had iASD (Table 3). Among them, 9 were detected by TEE and 15 by TTE. The total incidence of iASD was comparable between the two groups (21.4% vs. 15.4%, *p* = 0.429). All the patients with a patent iASD had LR shunts. The mean size of the patent iASD in the group 1 patients was 2.8 ± 0.9 [range: 1.5–5.1] mm, which was numerically larger than the 2.2 ± 0.8 [range: 1.4–3.5] mm for the patients in group 2 (*p* = 0.065). A total of two (1.8%) patients in group 1 developed two patent iASDs: the sizes of the iASDs were 4.5 + 2.0 mm and 2.3 + 1.7 mm. New-onset pulmonary hypertension was detected in three patients; all of them were in group 1, and none of them had a patent iASD. Two patients died, one (ICE-guided, with iASD in the first month) from device-related thrombus and the other (ICE-guided, without iASD in the second month) from catastrophic heart failure with delayed cardiac tamponade. No other adverse events occurred.

### 3.4. Twelve Months FU

There were 25 patients with patent iASD who received repeat TTE checks on day 243, on average (IQR: 186–382), after LAAC (Table 3). iASD was still present in five patients in group 1. The closure rate of patent iASD was comparable between the groups (70.6% vs. 71.4%, *p* = 1.000). No RL shunts were detected. The size of the residual iASD for the patients in group 1 was 2, 2, 3, 3, and 4 mm, and it was 1.5 and 2 mm for the patients in group 2 (*p* = 0.190). One patient in group 1 had two 4.0- and 1.0-mm shunts. The iASD occlusion of the patent is shown in Figure 3. None of the patients developed new-onset pulmonary hypertension. The data for all the patients who were alive after the first FU were available, and none of the patients died or developed adverse events.

### 3.5. Relationship between Patent iASD and ICE-Guided LAAC

Univariable logistic regression showed that ICE-guided LAAC did not increase the risk of iASD (Figure 4). However, both the history of ischemic stroke (OR = 2.488; 95% CI, 1.159–5.340; *p* = 0.019) and the use of the LACbes device (OR = 2.401; 95% CI, 1.005–5.737; *p* = 0.049) significantly increased the risk of iASD.

Even after adjustment for the CHA_2_DS_2_-VASc and HASBLED scores, we found no relationship between ICE-guided LAAC and the development of iASD (adjusted OR: 1.588 (95%CI, 0.696–3.623; *p* = 0.272)). We also found no relationship between ICE-guided LAAC and the development of iASD after adjustment for demographic characteristics (age, sex, and BMI); certain underlying diseases (non-paroxysmal AF, AF duration, heart failure, hypertension, diabetes mellitus, prior ischemic stroke, and coronary heart disease); the echocardiographic index (LAD, LVEF, pulmonary hypertension, pathological mitral, and tricuspid regurgitation); and peri-procedure characteristics (PFO closure and the use of different devices) (adjusted OR: 1.681 (95% CI, 0.634–4.455; *p* = 0.296)).

## 4. Discussion

The present study revealed the incidence, size, and clinical outcomes of iASDs after the ICE-guided LAAC procedure. The incidence of iASDs at month 2 after the ICE-guided LAAC procedure was 21.4%. The occlusion rate of iASDs at month 12 after the ICE-guided LAAC was 70.6%. There was no difference in the incidence of iASDs between the ICE-guided LAAC group and the TEE- or DSA-guided LAAC group at month 2 and 12. However, the size of iASD was numerically larger in the ICE-guided LAAC group than in the TEE- or ICE-guided LAAC group. Univariate and multivariable logistic regression analyses showed that the ICE-guided LAAC did not increase the risk of iASD. All the iASDs were LR shunt and did not cause pulmonary hypertension. The residual iASD was not associated with the development of adverse events. We found that both the history of ischemic stroke and the use of the LACbes device increased the risk of patent iASD, which, to the best of our knowledge, was never reported in the previous studies. To our knowledge, this is the first report on the dynamic incidence of patent iASD after ICE-guided LAAC.

### 4.1. Incidence and Size of Patent iASD after LAAC

The incidence of iASD shortly after LAAC has been reported in several studies. For instance, Korsholm et al. compared the efficacy and safety of ICE-guided LAAC and the traditional TEE-guided LAAC of the left upper chamber, with the iASD being one of the secondary efficacy parameters. The incidence and size of the residual iASD in the ICE group at day 55 of FU were 35% and 3.5 (IQR: 1–8) mm, respectively. For the TEE group, the incidence and size of the residual iASD were 26% and 3.5 (2–6) mm, respectively, which were not significantly different between the groups (*p* = 0.21 for the incidence of iASD and *p* = 0.58 for the iASD size) [8]. Puga et al. conducted single-center research on the risk of developing iASD. Thirty patients underwent ICE-guided LAAC on the left atrium, and 36 underwent TEE-guided LAAC. The results showed that the iASD incidence was higher in the ICE group (65.4% in the ICE group vs. 40.9% in the TEE group, *p* = 0.048), but there was no significant difference in the size of the iASD between the two groups (5 mm (IQR 4–5.5 mm) vs. 5 mm (2–5 mm), *p* = 0.712) at the first FU. Multivariable analysis showed that ICE alone was associated with a higher iASD (OR = 3.745, 95%CI: 1.197–11.715) incidence [9]. In the present study, we determined the incidence of iASD at month 2 of FU after LAAC. The results showed that the ICE-guided LAAC did not increase the risk of iASD. However, the size of the iASD was numerically larger in the ICE-guided LAAC group. Compared to the above reports, the incidence of iASD in our study was relatively low, and the size of the iASD was small. The mean iASD in the ICE group in this study was only 2.8 mm. For the long-term occlusion rate, Chen et al. reported that the incidence of iASD after the ICE-guided LAAC procedure dropped from 57.9% at month 2 of FU to 4.2% at month 12 of FU [12]. In our study, the incidence of iASD in the ICE-guided LAAC group was 21.4% at month 2 of FU, and the occlusion rate of iASD at month 12 of FU was 70.6%. Nelles et al. reported that the incidence of iASD under TEE-guided LAAC dropped from 34.7% at month 3 of FU to 18.1% at month 12 [13]. Singh et al. reported that the incidence of iASD in 253 patients in a “PROTECT AF” cohort at month 12 of FU was 6.8% [14]. The occlusion of iASD over time is a common phenomenon. To sum up, the short-term and long-term incidence and size of iASD following ICE-guided LAAC in our study are within the acceptable range.

### 4.2. Predictive Indicators of iASD

It is necessary to discuss the lack of difference between the ICE-guided group and the TEE- or DSA-guided group. The occurrence of iASD is related to many factors, such as sheath diameter, excessive operations, and FU duration. A sheath diameter ≥ 14-F increased the risk of iASD [15]. It was worth noting that the device delivery sheath was 12-F, and the sheath of the ICE catheter was 11-F, which meant that that the largest diameter of the puncture hole was almost 9 mm. However, our results showed that ICE-guided LAAC did not increase the risk of developing iASD, and the mean size of the patent iASD was 2.8 mm. During the procedure, the device delivery sheath and ICE catheter were advanced into the left atrium through the same puncture hole. Because the posterior and inferior area of the fossa ovalis was thick and elastic, we hypothesized that a further additional TP operation could not substantially damage the atrial septum. Two patients in group 1 had two patent iASDs at month 2 of FU, which might be related to extra damage caused by the poor operation. In addition, empirically speaking, ICE was more accurate in finding the device delivery pathway best suited to the LAAC procedure. Operators could look for an ideal puncture site using ICE images and the left atrial model. However, for the TEE-guided LAAC procedure, the primary use of the TEE images in TP was for safety purposes. The significant deviation between the axial direction of the device delivery catheter and the LAA inevitably increases the number of catheter operations, or there could even be a need re-perform the TP operation. Thirdly, iASD had a high spontaneous closure rate. The earlier the FU, the easier it was to observe the patent iASD. Puga et al. also reported that the ICE-guided approach was associated with a higher incidence of iASD. Their scheduled FU was quite early, even before the LAAC review, and the long-term occlusion rate was not reported [9]. To sum up, the ICE-guided approach did not increase the incidence of patent iASD.

We found that a history of ischemic stroke and the use of the LACbes device increased the risk of patent iASD. A relative observational study revealed that the cauliflower-shaped LAA was complex and was associated with cardiac embolism [16]. This kind of LAA would increase the difficulty of the operation, resulting in a higher number of LAAC attempts. In theory, because of the same sheath size, the use of the LACbes device does not increase the risk of iASD. In our study, the LACbes device was implanted if the LAA morphology was unsuitable for the WATCHMAN device implantation. Once the patient received the LACbes device implantation, he or she experienced a higher number of LAAC attempts compared to the patients who received the WATCHMAN device implantation. Excessive operations could increase the risk of iASD. A long-term observation study revealed that a higher number of cryoballoon (CBA) applications was an independent predictor of patent iASD (OR = 1.207, 95%CI, 1.033–1.411, *p* = 0.018) [17]. To minimize iASD development, a more precise evaluation before and during LAAC is necessary to improve the first successful deployment rate and minimize the TP attempts.

### 4.3. Clinical Outcomes of Patent iASD

The clinical relevance of patent iASD is still controversial. The previous studies show that blood flows through the LAAC-related iASD from the left to the right shunt, and the iASD resolves itself over time. In the present concept, the iASD was seen as a phenomenon rather than a complication, and the subsequent exclusive closure operation was rarely performed. Reports on the outcomes of iASDs after LAAC are scanty, but several studies have reported immediate or long-term iASDs after MitraClip. Schueler et al. reviewed 66 patients who underwent the MitraClip procedure and found that the iASD was the only indicator associated with death in 6 months [18]. Morikawa et al. investigated the outcomes of RL shunt through iASD and found that although RL shunt was only observed in 5% of the patients, it significantly increased the risk of acute deoxygenation. Moreover, the composite of major adverse cardiac events was significantly higher in the patients with RL shunt than those with LR shunt. By comparison, the patients with severe tricuspid regurgitation, higher serum B-type natriuretic peptide, higher pulmonary artery pressure, and right atrial pressure increased the risk of RL shunt [10]. For our cohort, cardiac function impairment was only mild; 39.7% of the patients presented with left heart failure; the mean LVEF was 55.4 ± 6.2%; the rate of pathological valve regurgitation at the baseline was less than 40%; none of the patients was in New York Heart Association functional class IV; and none of the patients had RL shunt through iASD over the FU period. It is worth noting that LAAC is appropriate for patients who have intolerance to anticoagulants or have a high risk of thromboembolism rather than those with severe hemodynamic disturbance. In addition, lifelong antithrombotic medications are recommended for patients who have undergone LAAC. In addition, the sheath diameter of the MitraClip catheter is 22-F, significantly larger than the 12-F LAAC catheter; a larger catheter can cause larger defects when punctured. Most studies have shown that LAAC-related iASD is not life-threatening and is not associated with adverse clinical events [9,11,12,13]. We believe that it is unnecessary to focus too much on LAAC-related iASD, whether TEE-, DSA-, or ICE-guided. Rather, there is a need to focus on better heart failure management to prevent iASD-associated complications.

### 4.4. Study Limitations

There were several limitations in our study. The rate of patent iASD was numerically higher in the ICE-guided LAAC group than in the TEE- or DSA-guided LAAC group. Considering that significantly more patients in the ICE-guided LAAC group than in the TEE- or DSA-guided LAAC group underwent RFCA in the same period of LAAC, we could not determine whether this difference was related to the insufficient sample size or was caused by the additional RFCA procedure. The sensitivity of TTE in detecting iASD is lower than that of TEE, which may lead to an underestimation of iASD. Given that this was a retrospective study, the accuracy of some data, such as the prevalence of migraine, might be inaccurate.

## 5. Conclusions

In the present study, the incidence of iASD following ICE-guided LAAC at month 2 of FU was 21.4%, and the 12-month closure rate of iASD was 70.6%. No patient had RL shunt. ICE-guided LAAC did not increase the risk of patent iASD. Despite the size of the iASD being numerically larger for the patient who underwent ICE-guided LAAC than the patient who underwent TEE- or DSA-guided LAAC, this did not increase the risk of developing pulmonary hypertension and other adverse clinical complications.

## Figures and Tables

**Figure 1 jcm-12-00160-f001:**
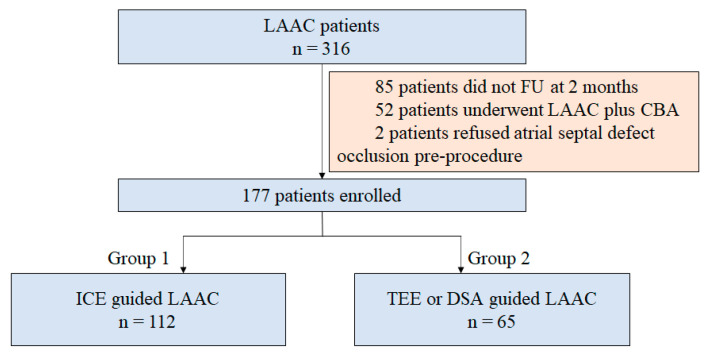
Study flowchart. LAAC = left atrial appendage closure; FU = follow-up; CBA = cryoballoon ablation; ICE = intracardiac echocardiography; TEE = transesophageal echocardiography; DSA = digital subtraction angiography.

**Figure 2 jcm-12-00160-f002:**
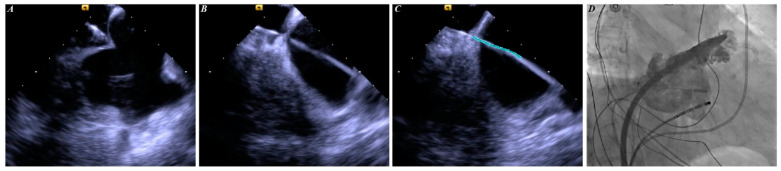
Steps of ICE-guided TP procedure. Panel (**A**) shows ICE image of tenting sign. Panel (**B**) shows ICE image of SL1 sheath positioned in the left atrium. Panel (**C**) shows delivered guide wire (green line) into the left atrium along the SL1 sheath. Panel (**D**) shows that SL1 sheath and guide wire were replaced, and device delivery sheath and ICE catheter were positioned in the left atrium through the same puncture hole. ICE = intracardiac echocardiography; TP = transseptal puncture.

**Figure 3 jcm-12-00160-f003:**
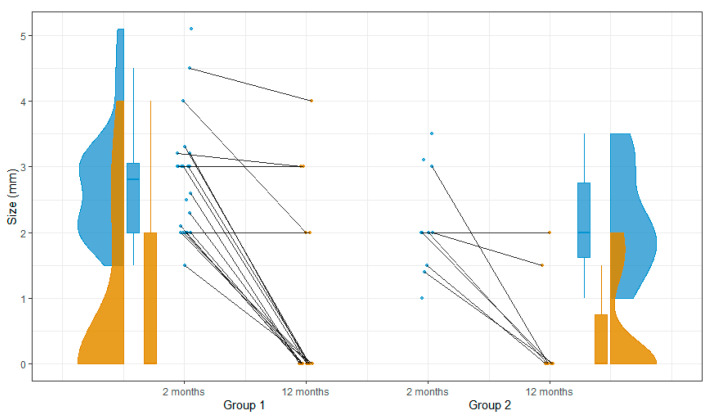
Occlusion of patent iASD. iASD = iatrogenic atrial septal defect.

**Figure 4 jcm-12-00160-f004:**
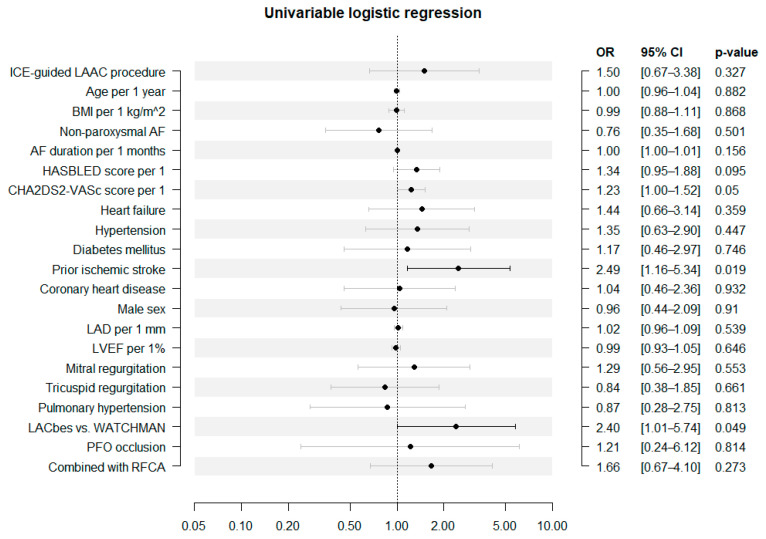
Predictors of patent iatrogenic atrial septal defect. ICE = intracardiac echocardiography; LAAC = left atrial appendage closure; BMI = body mass index; AF = atrial fibrillation; LAD = left atrial diameter; LVEF = left ventricular ejection fraction; PFO = patent foramen ovale; RFCA = radiofrequency catheter ablation.

**Table 1 jcm-12-00160-t001:** Baseline characteristics.

	Group 1 (*n* = 112)	Group 2 (*n* = 65)	*p*-Value
Demographics			
Age, years	*n* = 112, 64.6 ± 8.8	*n* = 65, 64.4 ± 9.6	0.844
Male sex	*n* = 112, 72 (64.3)	*n* = 65, 44 (67.7)	0.743
BMI, kg/m^2^	*n* = 112, 25.1 ± 3.2	*n* = 65, 25.0 ± 3.5	0.783
AF overview			
Paroxysmal AF	*n* = 112, 43 (38.4)	*n* = 65, 11 (16.9)	0.004
Non-paroxysmal AF	*n* = 112, 69 (61.6)	*n* = 65, 54 (83.1)	0.004
AF duration, months	*n* = 112, 12.0 (4.0, 48.0)	*n* = 65, 24.0 (6.0, 90.0)	0.041
Risk factors			
CHA_2_DS_2_-VASc score	*n* = 112, 3.3 ± 1.7	*n* = 65, 3.5 ± 1.9	0.551
Heart failure	*n* = 112, 34 (30.4)	*n* = 65, 22 (33.8)	0.738
Hypertension	*n* = 112, 58 (51.8)	*n* = 65, 41 (63.1)	0.160
Age ≥ 75 years	*n* = 112, 11 (9.8)	*n* = 65, 9 (13.8)	0.464
Stroke/TIA/SE	*n* = 112, 39 (34.8)	*n* = 65, 19 (29.2)	0.508
Coronary artery disease	*n* = 112, 32 (28.6)	*n* = 65, 19 (29.2)	1.000
Age 65–74 years	*n* = 112, 50 (44.6)	*n* = 65, 25 (38.5)	0.435
HASBLED score	*n* = 112, 1.8 ± 1.1	*n* = 65, 1.8 ± 1.2	0.812
Echocardiographic index			
LAD, mm	*n* = 112, 44.6 ± 5.6	*n* = 65, 46.1 ± 6.7	0.120
LVEF, %	*n* = 112, 55.4 ± 6.0	*n* = 65, 55.3 ± 5.8	0.886
Pathological mitral regurgitation	*n* = 112, 28 (25.0)	*n* = 65, 17 (26.2)	0.860
Pathological tricuspid regurgitation	*n* = 112, 37 (33.0)	*n* = 65, 26 (40.0)	0.416
Pulmonary hypertension	*n* = 112, 7 (6.3)	*n* = 65, 16 (24.6)	0.001
Mean pressure, mmHg	*n* = 7, 45.0 ± 3.5	*n* = 16, 43.6 ± 8.6	0.678

BMI = body mass index; AF = atrial fibrillation; TIA = transient ischemic attack; SE = systemic embolism; LAD = left atrial diameter; LVEF = left ventricular ejection fraction.

**Table 2 jcm-12-00160-t002:** Procedure characteristics.

	Group 1 (*n* = 112)	Group 2 (*n* = 65)	*p*-Value
Device			0.414
WATCHMAN	*n* = 112, 90 (80.4)	*n* = 65, 55 (84.6)	
LACbes	*n* = 112, 22 (19.6)	*n* = 65, 9 (13.8)	
Intra-procedure details			
LAAC success	*n* = 112, 111 (99.1)	*n* = 65, 64 (98.5)	1.000
Combined with RFCA	*n* = 112, 101 (90.2)	*n* = 65, 26 (40.0)	0.000
PFO occlusion	*n* = 112, 0 (0.0)	*n* = 65, 9 (13.8)	0.000
Discharge medication			0.011
OAC monotherapy	*n* = 112, 107 (95.5)	*n* = 65, 57 (87.7)	
OAC plus single antiplatelet therapy	*n* = 112, 5 (4.5)	*n* = 65, 3 (4.6)	
Dual antiplatelet therapy	*n* = 112, 0 (0.0)	*n* = 65, 5 (7.7)	

LAAC = left atrial appendage closure; RFCA = radiofrequency catheter ablation; PFO = patent foramen ovale; OAC = oral anticoagulant.

**Table 3 jcm-12-00160-t003:** Follow-up.

	Group 1 (*n* = 112)	Group 2 (*n* = 65)	*p*-Value
First FU (month 2)			
Time to review, days	*n* = 112, 57.5 (44.0, 75.0)	*n* = 65, 57.0 (50.0, 71.0)	0.236
Examination			0.001
TEE	*n* = 112, 33 (29.5)	*n* = 65, 35 (53.8)	
TTE	*n* = 112, 79 (70.5)	*n* = 65, 30 (46.2)	
Patent iASD	*n* = 112, 24 (21.4)	*n* = 65, 10 (15.4)	0.429
TEE-detected	*n* = 33, 9 (27.3)	*n* = 35, 4 (11.4)	0.128
TTE-detected	*n* = 79, 15 (19.0)	*n* = 30, 6 (20.0)	1.000
Shunt			1.000
LR shunt	*n* = 24, 24 (100.0)	*n* = 10, 10 (100.0)	
RL shunt	*n* = 24, 0 (0.0)	*n* = 10, 0 (0.0)	
Size, mm	*n* = 24, 2.8 ± 0.9	*n* = 10, 2.2 ± 0.8	0.065
New-onset pulmonary hypertension	*n* = 112, 3 (2.68)	*n* = 65, 0 (0.0)	0.299
Final FU (month 12)			
Time to review, days	*n* = 17, 221.0 (161.0, 392.0)	*n* = 7, 260.0 (204.0, 364.0)	0.619
Residual iASD	*n* = 17, 5 (29.4)	*n* = 7, 2 (28.6)	1.000
Shunt			1.000
LR shunt	*n* = 5, 5 (100.0)	*n* = 2, 2 (100.0)	
RL shunt	*n* = 5, 0 (0.0)	*n* = 2, 0 (0.0)	

FU = follow-up; TEE = transesophageal echocardiography; TTE = transthoracic echocardiography; iASD = iatrogenic atrial septal defect; LR = left-to-right; RL = right-to-left.

## Data Availability

The datasets used and/or analyzed during the current study are be available from the corresponding author on reasonable request.

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
