# Peer review of "Iatrogenic Atrial Septal Defect after Intracardiac Echocardiography-Guided Left Atrial Appendage Closure: Incidence, Size, and Clinical Outcomes"

_jcm, 2022, doi:10.3390/jcm12010160_

Round 1

Reviewer 1 Report

I’ve read with great interest the paper from Yi-Bo Ma, Lan-Yan Guo et al who investigate the incidence, size, and clinical outcomes of iASD after ICE-guided LAAC.

In this single-center retrospective study, they enrolled 177 patients were divided in:

-        ICE guided LAAC (112 pts)

-        TEE or DSA guided LAAC (65 pts)

The two groups are well-matched despite there were not randomized.

As expected no differences in results were found.

The final message is simple and practical however too many tables and results are shown in the manuscript.

This will make the manuscript very complex when it should flow quickly.

The discussion does not clarify the aim of the study and does not focus on results analysis. The authors use the discussion session for a detailed comparison with previous data presented in the literature. 

Results and discussion should be reconsidered.

Author Response

Dear professor:

I am very grateful to your comments for the manuscript. According with your advice, we amended the relevant part in manuscript. Some of your questions were answered below.

Response to Reviewer 1 Comments

Point 1: The final message is simple and practical however too many tables and results are shown in the manuscript. This will make the manuscript very complex when it should flow quickly.

Response 1: Thank you very much for your valuable suggestions. We have amended the part of Results. Patient Characteristics At Baseline and Procedure Characteristics were divided into two separate paragraphs firstly. We also delete the unimportant table (Table 4) and unimportant details in the tables. Whether the details are important depends on whether they follow the logic of the article. Although we did not observe iASD-related complications, many baseline and procedure characteristics did affect the risk of adverse events, such as antithrombotic therapy. These characteristics were necessary to show in the manuscript. The nature of this study was retrospective, and multivariable regression was necessary. This was extremely important to rule out the interference of confounding factors. We also wanted to explore the potential risk factors of iASD by univariable regression analyses before multivariable analysis. We might be able to learn from the potential risk factors whether the increase of iASD was due to the patient himself or the operations. The above is the logic of this article. In present manuscript, each paragraph is corresponded to only one table or figure. We believe this will make the manuscript more structured.

Point 2: The discussion does not clarify the aim of the study and does not focus on results analysis. The authors use the discussion session for a detailed comparison with previous data presented in the literature.

Response 2: Originally, the first part of Discussion aimed to briefly summarize the findings in our study; the second part aimed to described and compared the incidence and size of iASD through systemically reviewing previous reports; the third part was used to analyze the difference of iASD between the two groups and analyze the potential risk factors; the fourth part aimed to analysis the adverse events of iASD. As you commented, there are some problems with our discussion. We did not clarify the aim of the study. And we did not pay more attention to results analysis. Now, we emphasized the main purpose and findings of the article in the first part of Discussion. Besides, we enrich the results analysis of iASD differences between the two groups in the third part. Thank you very much for your comment.

Best regards,

Yibo Ma

Reviewer 2 Report

Line 37 – I suggest adding as Reference - Guidelines for AF treatment (ESC Guidelines, or AHA Guidelines, or even local Guidelines).

Line 63 – Authors define the study as RETROSPECTIVE! The way they describe it in Study Population it seems as a Prospective study– written inform consent, follow up period etc…

Line 100- I suggest to add few lines about Group 2 - TEE or DSA-guided group,

Line 157, 174 – I suggest naming of FUs – to be 2 months FU, and 12 months FU, instead of first and last FU.

 Line 274 – “Sheath The clinical relevance”- I suppose it is technical mistake.  

Author Response

Dear professor:

I am very grateful to your comments for the manuscript. According with your advice, we amended the relevant part in manuscript. Some of your questions were answered below.

Response to Reviewer 2 Comments

Point 1: Line 37 – I suggest adding as Reference - Guidelines for AF treatment (ESC Guidelines, or AHA Guidelines, or even local Guidelines).

Response 1: We have added the 2020 ESC Guidelines (2020 ESC Guidelines for the diagnosis and management of atrial fibrillation developed in collaboration with the European Association for Cardio-Thoracic Surgery (EACTS)) (Line 37).

Point 2: Line 63 – Authors define the study as RETROSPECTIVE! The way they describe it in Study Population it seems as a Prospective study– written inform consent, follow up period etc…

Response 2: We have amended the description of “Study population” and “Outpatient follow up” (Line 69, 78, 114 and 118).

Point 3: Line 100- I suggest to add few lines about Group 2 - TEE or DSA-guided group.

Response 3: We have added the brief description of TEE or DSA-guided TP procedure (Line 105 – 112). Thank you very much for you valuable comment.

Point 4: Line 157, 174 – I suggest naming of FUs – to be 2 months FU, and 12 months FU, instead of first and last FU.

Response 4: We have renamed the first FU and last FU to Two months FU and Twelve months FU, respectively (Line 163 and 180).

Point 5: Line 274 – “Sheath The clinical relevance”- I suppose it is technical mistake.

Response 5: This is a technical mistake. The sentence should be “The clinical relevance ...” .

Best regards,

Yibo Ma

Reviewer 3 Report

Dear Sir/Madam,

I had the opportunity to act as a reviewer on the recent submission by Ma et al. to the Journal of Clinical Medicine.

The authors present an interesting paper studying incidence, size, and clinical outcomes of iatrogenic atrial septal defect after ICE-guided left atrial appendage closure. The key finding is that ICE-guided left atrial appendage closure does not increase the risk of iatrogenic atrial septal defect. 

The manuscript is very well structured and written. However, some issues need to be addressed: 

1.     Please describe what devices were exactly used in all groups. In the case of two or more devices please provide the exact dimensions of the delivery sheaths and comment on the findings from this perspective.

2.     What exactly is the meaning of adverse events? (line 183).

Best regards,

Author Response

Dear professor:

I am very grateful to your comments for the manuscript. According with your advice, we amended the relevant part in manuscript. Some of your questions were answered below.

Response to Reviewer 3 Comments

Point 1: Please describe what devices were exactly used in all groups. In the case of two or more devices please provide the exact dimensions of the delivery sheaths and comment on the findings from this perspective.

Response 1: Both WATCHMAN device (Boston Scientific) and LACbes device (PushMed, Shanghai, China) were used in all groups. Both sheaths have a diameter of 12-F. In theory, the use of LACbes device does not increase the incidence of iASD. But we found that the use of LACbes device significantly increased the risk of iASD (OR = 2.401; 95% CI, 1.005 – 5.737; p = 0.049). LACbes device was instead implanted if the LAA morphology was unsuitable for WATCHMAN device implantation. We believe that the use of LACbes device represented excessive operations. Relevant results have been amended. Thank you very much for your professional comments.

Point 2: What exactly is the meaning of adverse events?

Response 2: Adverse events in our study included cardiovascular death, ischemic stroke, migraine, and readmission for acute heart failure. New-onset pulmonary hypertension was not only an echocardiographic outcome, but also an adverse event. During follow-up, excepted for two patients died from iASD-unrelated complications at month 2, we did not find other events. Patients with patent or residual iASD did not experienced new-onset pulmonary hypertension.

Best regards,

Yibo Ma
